# Deepening the Mechanisms of Visceral Pain Persistence: An Evaluation of the Gut-Spinal Cord Relationship

**DOI:** 10.3390/cells9081772

**Published:** 2020-07-24

**Authors:** Elena Lucarini, Carmen Parisio, Jacopo J. V. Branca, Cristina Segnani, Chiara Ippolito, Carolina Pellegrini, Luca Antonioli, Matteo Fornai, Laura Micheli, Alessandra Pacini, Nunzia Bernardini, Corrado Blandizzi, Carla Ghelardini, Lorenzo Di Cesare Mannelli

**Affiliations:** 1Department of Neuroscience, Psychology, Drug Research and Child Health, Neurofarba, Pharmacology and Toxicology Section, University of Florence, Viale Pieraccini 6, 50139 Florence, Italy; elena.lucarini@unifi.it (E.L.); carmen.parisio@unifi.it (C.P.); laura.micheli@unifi.it (L.M.); carla.ghelardini@unifi.it (C.G.); 2Department of Experimental and Clinical Medicine—DMSC, Anatomy and Histology Section, University of Florence, L. go Brambilla 3, 50134 Florence, Italy; jacopojuniovalerio.branca@unifi.it (J.J.V.B.); alessandra.pacini@unifi.it (A.P.); 3Department of Clinical and Experimental Medicine, Unit of Histology, University of Pisa, 56126 Pisa, Italy; cristina.segnani@unipi.it (C.S.); chiara.ippolito@unipi.it (C.I.); nunzia.bernardini@med.unipi.it (N.B.); 4Department of Pharmacy, Unit of Pharmacology, University of Pisa, 56126 Pisa, Italy; carolina.pellegrini@unipi.it; 5Department of Clinical and Experimental Medicine, Unit of Pharmacology and Pharmacovigilance, University of Pisa, 56126 Pisa, Italy; luca.antonioli@medmcs.unipi.it (L.A.); matteo.fornai@unipi.it (M.F.); corrado.blandizzi@unipi.it (C.B.); 6Interdepartmental Research Center “Nutraceuticals and Food for Health”, University of Pisa, 56126 Pisa, Italy

**Keywords:** visceral hypersensitivity, inflammatory bowel disease, irritable bowel syndrome, microglia, astrocytes, immune system

## Abstract

The management of visceral pain is a major clinical problem in patients affected by gastrointestinal disorders. The poor knowledge about pain chronicization mechanisms prompted us to study the functional and morphological alterations of the gut and nervous system in the animal model of persistent visceral pain caused by 2,4-dinitrobenzenesulfonic acid (DNBS). This agent, injected intrarectally, induced a colonic inflammation peaking on day 3 and remitting progressively from day 7. In concomitance with bowel inflammation, the animals developed visceral hypersensitivity, which persisted after colitis remission for up to three months. On day 14, the administration of pain-relieving drugs (injected intraperitoneally and intrathecally) revealed a mixed nociceptive, inflammatory and neuropathic pain originating from both the peripheral and central nervous system. At this time point, the colonic histological analysis highlighted a partial restitution of the *tunica mucosa*, transmural collagen deposition, infiltration of mast cells and eosinophils, and upregulation of substance P (SP)-positive nerve fibers, which were surrounded by eosinophils and MHC-II-positive macrophages. A significant activation of microglia and astrocytes was observed in the dorsal and ventral horns of spinal cord. These results suggest that the persistence of visceral pain induced by colitis results from maladaptive plasticity of the enteric, peripheral and central nervous systems.

## 1. Introduction

Chronic and recurrent abdominal pain is commonly experienced in the general population, including children and adolescents. Despite the high prevalence, there is a major gap in the knowledge of pathogenic mechanisms underlying chronic pain syndromes, thereby reducing strongly the possibility of identifying effective therapeutic interventions [1].

Abdominal pain frequently results from prolonged enteric inflammatory processes, as in cases of inflammatory bowel diseases (IBDs) [2,3]. Bowel inflammation leads to enteric barrier breakdown, altered water/electrolyte secretion, changes in motility patterns, and visceral sensations, culminating in abdominal symptoms (diarrhea, cramping, and pain) [4]. Although pain in IBDs is traditionally attributed to the severity of inflammation, 20–50% of patients complain of a disabling abdominal pain also during the periods of clinical and endoscopic remission [3,4]. The poor correlation between reported abdominal pain intensity and IBD activity raises close similarities with irritable bowel syndrome (IBS) [2,5], a symptom-based clinical condition defined by the persistence of abdominal pain and discomfort, with altered bowel habits, in the absence of any other disease [6]. The development of IBS is frequent after severe infections or prolonged inflammatory processes [6]. The current therapeutic approaches to IBD and IBS symptoms, including psycho-nutritional and pharmacological treatments (bulking-agents, antidiarrheals, antispasmodics, antidepressants) [4,7], are almost ineffective against abdominal pain [8] which highly impacts the patient’s quality of life.

The development of novel effective therapies needs the elucidation of mechanisms involved in the development and persistence of visceral pain. Since different neuro-immuno-biological pathways within the bowel and nervous system (including enteric, peripheral, and central compartments) contribute to visceral pain perception [2,4,9,10], the present study was designed to pursue the following aims: (i) to investigate the evolution of visceral hypersensitivity in the post-inflammatory phase of colitis; (ii) to analyze the pathophysiological features of abdominal pain by testing its response to drugs administered systemically or intrathecally; (iii) to examine the intestinal and spinal cord histo-morphological alterations, and their relationship in the persistence of pain. For these purposes, we employed the rat model of colitis induced by intra-rectal injection of 2,4-dinitrobenzenesulfonic acid (DNBS) [11], since these animals have been shown to display persistent visceral hypersensitivity and bowel dysmotility after the remission of intestinal inflammation [12,13].

## 2. Materials and Methods

### 2.1. Animals

For all the experiments described below, male Sprague–Dawley rats (Envigo, Varese, Italy), weighing approximately 220–250 g at the beginning of the experimental procedure, were used. Animals were housed in CeSAL (Centro Stabulazione Animali da Laboratorio, University of Florence, Florence, Italy) and used at least 1 week after their arrival. Four rats were housed per cage (size 26 × 41 cm); animals were fed a standard laboratory diet and tap water ad libitum, and kept at 23 ± 1 °C with a 12 h light/dark cycle, light at 7 a.m. All animal manipulations were carried out according to the Directive 2010/63/EU of the European Parliament and the European Union Council (22 September 2010) on the protection of animals employed for scientific purposes. The ethical policy of the University of Florence complies with the Guide for the Care and Use of Laboratory Animals of the US National Institutes of Health (NIH Publication number 85–23, revised 1996, University of Florence assurance number: A5278-01). Formal approval to conduct the described experiments was obtained from the Animal Subjects Review Board of the University of Florence (543/2017-PR). Experiments involving animals have been reported according to ARRIVE guidelines [14]. All efforts were made to minimize animal suffering and to reduce the number of employed animals.

### 2.2. Induction of Colitis

Colitis was induced as previously described [11], through an intrarectal injection of 2,4-dinitrobenzenesulfonic acid DNBS (30 mg in 0.25 mL of ethanol 50%). Control rats received 0.25 mL of saline solution.

### 2.3. Drug Administrations

Morphine hydrochloride (S.A.L.A.R.S., Como, Italy) was administered subcutaneously and intrathecally [15,16]; ibuprofen [17] and otilonium bromide [18] (Carbosynth, Compton, UK) were administered orally. Pregabalin [17,19] and amitryptiline hydrochloride [20] (Sigma-Aldrich, Milan, Italy) were administered intraperitoneally and intrathecally. Dexamethasone [21] (Carbosynth, Compton, UK) was administered intraperitoneally. Transcutaneous intrathecal injections were performed under anesthesia (isoflurane 2%); the compounds were dissolved in phosphate-buffered saline and administered in a final volume of 15 µL, as previously described by Mestre et al. [22]. All the compounds were administered 15 min before tests.

### 2.4. Assessment of Visceral Sensitivity by Viscero-Motor Response and Abdominal Withdrawal Reflex

For the electromyographic (EMG) recordings, in animals under anesthesia (isoflurane 2%) two electrodes (AS631, Cooner Wire, Chatsworth, CA, USA) were sewn into the external oblique abdominal muscle and exteriorized dorsally as previously described by Christianson and Gebhart (2007) [23] with minor changes to adapt it to rats. The viscero-motor response (VMR) to Colo-Rectal Distension (CRD) was recorded as previously reported in the animals under anesthesia (isoflurane 2%) [23,24]. To perform colo-rectal distension, a balloon (length: 4.5 cm) was inserted into the colon and filled with increasing volumes of water (0.5, 1, 2, 3 mL); 5 min was the time elapsed between two consecutive distensions). This paradigm was used for the evaluation of visceral sensitivity over time and for the assessment of drug efficacy. A smaller balloon (length: 2 cm) was used to perform rectal distension by its positioning in the rectum and filling it with 1.5 mL (this paradigm was applied to evaluate visceral sensitivity in a distal region of the gut rather than that affected directly by DNBS damage). During the test, the electrodes were connected to a data acquisition system and the corresponding EMG signals, consequent to colo-rectal stimulations were recorded, amplified, and filtered (Animal Bio Amp, ADInstruments, Colorado Springs, CO, USA), digitized (PowerLab 4/35, ADIinstruments), analyzed, and quantified using LabChart 8 (ADInstruments). To quantify the magnitude of the viscero-motor response at each distension volume, the area under the curve (AUC) immediately preceding the distension (30 s) was subtracted from the AUC during the balloon distension (30 s), and the responses were expressed as percent increments from the baseline. Behavioral responses to colo-rectal distension were assessed via abdominal withdrawal reflex (AWR) measurement in conscious animals by a semi-quantitative score, as described previously [25]. Briefly, rats were anesthetized with isoflurane, and a lubricated latex balloon (length: 4.5 cm), attached to a polyethylene tubing, assembled to an embolectomy catheter, and connected to a syringe filled with water, was inserted through the anus into the rectum and descending colon. The tubing was taped to the tail to hold the balloon in place. Then rats were allowed to recover from anesthesia for 30 min. AWR measurement consisted of visual observations of animal responses to graded CRD (0.5, 1, 2, 3 mL) by blinded observers, who assigned the following AWR scores: no behavioral response to colo-rectal distention (0); immobile during colorectal distention and occasional head clinching at stimulus onset (1); mild contraction of the abdominal muscles but absence of abdomen lifting from the platform (2); strong contraction of the abdominal muscles and lifting of the abdomen off the platform (3); arching of the body and lifting of the pelvic structures and scrotum (4). The time elapsed between two consecutive distension was 5 min.

### 2.5. Assessment of Depression and Anxiety-Related Behaviors

Forced swimming test (FST) [26]: A pre-swim session (15 min) was conducted 24 h prior to the test. On the test day, the animals were introduced again into the cylinder filled with water (24 °C) to a depth of 30 cm: the mobility time and travelled distance within 5 min were the parameters of interest.

Open field test (OFT) [27]: Rats were placed into the center of an arena, brightly lit (1000 lux). The total distance travelled and the time spent by animals in the center of the arena within 5 min were recorded.

Elevated plus maze test (EPMT) [28]: Rats were placed in the center and allowed to explore the maze for 5 min. The percentages of time spent in the open arms, in the closed arms, and in the center were measured.

Test sessions were recorded by a video camera positioned directly above the dedicate apparatus. Data acquisition and analysis were performed automatically using ANY-maze Software^®^ (Stoelting Co., Chicago, IL, USA).

### 2.6. Histological, Histochemical and Immunoistological Analysis of the Colon

The evaluation of colonic macroscopic damage was performed in accordance with the criteria and score reported previously [11]: presence of adhesions between colon and other intra-abdominal organs (0–2); consistency of colonic fecal material (0–2); thickening of colonic wall (mm); presence and extension of hyperemia and macroscopic mucosal damage (0–5). Microscopic evaluations were carried out on hematoxylin/eosin-stained sections of full-thickness samples obtained from the distal colon, which were 4% paraformaldehyde fixed, paraffin embedded, and sectioned in 7 µm slices. The microscopic damage was scored in accordance with the criteria reported previously [29]: mucosal architecture loss (0–3); goblet cell depletion (0, absent; 1, present); crypt abscess (0, absent; 1, present); cellular infiltration (0–3); *tunica muscularis* thickening (0–3).

Collagen deposition was assessed by the Sirius red/fast green (SR/FG) staining. Mast cells (MCs) and eosinophils were detected by specific histochemical staining carried out with toluidine blue and hematoxylin/eosin staining, respectively, as reported by Zhao et al. [30]. Substance P (SP)-positive nerve fibers were detected by immunoenzymatic histochemistry employing validated immunoglobulins specific for rat central neurons [31]. Sections were examined by a Leica DMRB light microscope, equipped with a DFC480 digital camera (Leica Microsystems, Mannheim, Germany), and analyzed quantitatively using the Image Analysis System “L.A.S. software version 4.5” (Leica Microsystems, Mannheim, Germany). Two blind investigators evaluated independently the eosinophil and MC density (cell number/respective analyzed areas (mm^2^)) and the SR and SP-positivity expression (positive pixels percentage (PPP)) [32,33]. Activated macrophages were revealed by immunofluorescence staining of the MHC-II antigen [34], which is a determinant of the 1-A antigen present in rat intestinal macrophages [34,35], by specific immunoglobulins validated in rat spleen tissue and able to detect macrophages in the central nervous system [36]; their distribution with respect to SP-positive nerve fibers was examined on colonic sections by double immunofluorescence protocols under a Leica TCS SP8 confocal laser-scanning microscope (Leica Microsystems, Germany; for reagents see Appendix A) [37].

### 2.7. Immunofluorescence of the Spinal Cord

The lumbar segments of the rat spinal cord were exposed from the vertebral column via laminectomy and formalin-fixed by standard protocols as described previously [38,39]. Cryostat sections (5 μm thickness) were washed thrice with phosphate-buffered saline (PBS), permeabilized with 0.3% Triton X-100 in PBS (PBST) for 10 min, and then were incubated, at room temperature, for 1 h in blocking solution (1% bovine serum albumin in PBST). Slices were incubated overnight at 4 °C in PBST containing the appropriate primary antibodies. The primary antibody was directed against Iba1 (ionized calcium-binding adapter molecule 1) for microglial staining or against GFAP (glial fibrillary acidic protein) for astrocyte staining. The following day, slides were washed thrice with PBS, and then incubated in blocking solution for 1 h with goat anti-rabbit IgG secondary antibodies labeled with Alexa Fluor 488 for microglia and 568 for astrocytes. To stain nuclei, sections were incubated with DAPI in PBST for 5 min, at room temperature in the dark (for reagents see Appendix A). After three washes in PBS and a final wash in distilled water, slices were mounted using ProLong Gold (Life Technologies-ThermoFisher Scientific, Milan, Italy) as mounting medium. Digitalized images were collected at 100×, 200×, or 400× total magnification by a motorized Leica DM6000B microscope equipped with a DFC350FX. Quantitative analysis of GFAP- and Iba1-positive cells was performed by collecting independent fields in the dorsal and ventral horn of each spinal cord, by using the “cell counter” plugin of ImageJ (NIH, Bethesda, MD, USA).

A total of 16 spinal cord sections (8 in the dorsal and 8 in the ventral horn) for each animal were used for quantitative analysis. Each value represents the mean of 8 fields per spinal cord. Control group (*n* = 5), DNBS group (*n* = 5).

For each biomarker, primary antibodies were omitted for negative controls.

### 2.8. Statistics

Behavioral measurements were made by researchers blinded to animal treatments. Behavioral tests were conducted on different groups of animals. The same experimental group was used for evaluating animal behavior on multiple experimental days. All the results were expressed as means ± S.E.M. and the analysis of variance was performed by one or two-way ANOVA. With regard to behavior and histopathological assessments of the gut, a Bonferroni’s significant difference procedure was used for *post hoc* comparison. A *t*-test was performed for the statistical analysis of collagen fiber expression, MC and eosinophil density, SP immunostaining, and spinal cord immunofluorescence. *p* values lower than 0.05 or 0.01 were considered significant. Data were examined using the “Origin 9” software (OriginLab, Northampton, MA, USA).

## 3. Results

### 3.1. Assessment of Visceral Sensitivity

Visceral sensitivity was assessed by measuring the VMR to colo-rectal distension (0.5–3 mL) at baseline (pretest) as well as 7, 14, and 21 days after DNBS injection (Figure 1a). No differences were found in the experimental groups before treatment (Figure 1, pretest). The induction of colitis was associated with a remarkable and persistent increase in visceral sensitivity. DNBS-treated animals displayed a significantly higher VMR to colorectal distension, as compared to control animals up to 21 days after the injection (Figure 1a). On day 14, the DNBS group was responsive also to 1 mL stimulation, displaying a further lowering of the visceral sensitivity threshold (Figure 1a).

The VMR to a localized rectal stimulus (2 cm balloon filled with 1.5 mL) was evaluated. The response of DNBS-treated animals was significantly increased, as compared to controls at each time point (Figure 1b).

The behavioral nocifensive response of animals was measured by assigning a score to their AWR to colo-rectal distension (0.5–3 mL; Figure 1c). The score of vehicle-treated animals was constant over time and proportional to the applied stimulus. DNBS-treated animals displayed a significantly higher abdominal response than controls, being responsive also to low distension volume (0.5–1 mL). This effect was observed up to 91 days after DNBS injection (Figure 1c).

### 3.2. Assessment of Behavioral Alteration

The behavioral alterations frequently associated with the presence of spontaneous pain, such as depression, anxiety, and reduction of locomotor activity, were investigated through the FST, OFT, and EPMT. FST is used to highlight depressive-like behaviors in animals. Control rats swam, struggled, and climbed the wall of the tank to stay afloat in the water. This behavior was expressed both as the time the animals spent moving and as the travelled distance. Despite the changes observed over time within the experimental groups, DNBS-treated animals showed decreased mobility in comparison with controls either on days 7, 14, and 21, pointing out a persistent depressive-like behavior (Figure 1d). OFT was performed to evaluate the locomotor activity and explorative behavior (expressed as travelled distance and wall rearing, respectively). Both the travelled distance and the number of wall rearing performed by animals with colitis were significantly lower than that of controls 7 days after DNBS injection. By contrast, no differences were recorded between the experimental groups on days 14 and 21 (Figure 1e). The time spent in the corners of the arena did not differ among groups suggesting the lack of anxiety related phenomena. This result was confirmed by EPMT; indeed, DNBS-treated animals did not show significant differences in the percentages of time spent in the open arms, in the closed ones, and in the center with respect to controls. In both groups, the animals preferred to spend their time in the closed arms instead of the open arms (Figure 1f).

### 3.3. Histological Assessment of Colonic Damage

Groups of animals were sacrificed 3, 7, 14, and 21 days after DNBS injection, and each colon was harvested and processed for both macroscopic and microscopic analysis (Figure 2a,b, respectively). Macroscopically, the damage associated with colitis showed a peak three days after DNBS injection and decreased progressively from day 7 to day 21. On day 21, the macroscopic damage score of animals with colitis was still significantly higher than for controls, but considerably lowered with respect to that estimated on day 3 (Figure 2a). The microscopic damage score followed the same trend at the respective time points (3, 7, 14, and 21 days; Figure 2b,c).

Histological analysis revealed that colitis peaked three days after DNBS injection—the colon appeared extensively inflamed, infiltrated, and thickened with diffuse ulcerations and necrosis, and loss of its main layers. Seven days after DNBS injection, colonic samples still showed considerable epithelial injuries. The tissue largely recovered its primary structures, though there were several ulcers and areas with lining epithelium loss, transmural immune cell infiltration (predominantly neutrophils, lymphocytes, and MCs), crypt abscesses, altered goblet cells, and edema. On day 14, the colon appeared still significantly thickened with inflammatory infiltration. The *tunica mucosa* was mostly restored, apart from spot loss of the epithelial surface, probably resulting from healing processes on previous deep ulcers. The crypts were elongated with irregular diameters and shapes. On day 21, although the thickening of colonic wall persisted, the presence of inflammatory infiltrate was reduced and almost exclusively limited to the submucosa. No hyperplasia of epithelial cells was detected, and the structure of the crypts was comparable to that of the controls (Figure 2c).

Based on the above results, the optimal timing for evaluating persistent visceral hypersensitivity in the presence of a significant restoration of bowel morphology was established at day 14. Thus, the subsequent behavioral and histological evaluations of DNBS-treated animals were performed at this time point.

### 3.4. Effects of the Systemic Administration of Reference Drugs

To investigate the characteristics of DNBS-induced pain, we evaluated over time (7, 14, and 21 days after DNBS injection; Figure 3a–c, respectively) the effects of the acute systemic administration of different drugs on visceral hypersensitivity. Abdominal pain was assessed by measuring VMR to colo-rectal distension 15 min after drug administration (Figure 3). The acute administration of morphine (5–10 mg kg^−1^ s.c.) and amitriptyline (15 mg kg^−1^ i.p.) suppressed VMR completely in DNBS-treated animals 7, 14, and 21 days after colitis induction (Figure 3a–c). Otilonium bromide (20 mg kg^−1^ p.o.) significantly reduced visceral hypersensitivity on day 7 (Figure 3a), while it was less effective on days 14 and 21 (Figure 3b,c). By contrast, pregabalin (30 mg kg^−1^ i.p.), which was ineffective in reducing visceral hypersensitivity on day 7 (Figure 3), became increasingly active starting from day 14 (Figure 3b,c). Ibuprofen (100 mg kg^−1^ p.o.) was not able to reduce significantly visceral pain induced by DNBS (Figure 3a–c). Likewise, dexamethasone (0.6 mg kg^−1^ i.p.) was also ineffective (Figure 3a–c).

Considering the lack of substantial differences among the effect of drugs on day 14 and 21, hereinafter we report only the results gathered on day 14 as the representative phase of visceral pain persistence.

Figure 3d shows the effect of the acute systemic administration of the above drugs on AWR in DNBS-treated animals. The systemic administration of morphine abolished completely the DNBS-induced hypersensitivity, increasing the pain threshold even beyond controls. On the other hand, amitriptyline and otilonium bromide retained their efficacy on visceral hypersensitivity without affecting the basal threshold of intestinal sensitivity. Dexamethasone was ineffective also in this setting, while ibuprofen became partially effective. By contrast, pregabalin lost some of its effectiveness when evaluated in this behavioral paradigm (Figure 3d).

### 3.5. Effects of Intrathecal Administration of Reference Drugs

Figure 3e shows the effects of the intrathecal (i.t.) administration of morphine, amitriptyline, and pregabalin on both VMR (Figure 3e) and AWR (Figure 3f) to colorectal distension in DNBS-treated animals (day 14). Pregabalin (100 µg i.t.) strongly reduced VMR (Figure 3e), while being ineffective on AWR (Figure 3f). Amitriptyline (60 µg i.t.) exerted a partial effect on AWR (Figure 3f) without affecting VMR (Figure 3e). Likewise, morphine (1 µg i.t.) reduced the AWR significantly in DNBS-treated animals, without affecting VMR.

### 3.6. Histological Evaluation of Inflammatory Cells, Fibrosis, and SP-Immunostained Nerve Fibers

On day 14 after DNBS treatment, the partial restitution of colonic *tunica mucosa* was associated with inflammatory cell infiltration and significant transmural deposition of SR-positive collagen fibers (Figure 4a). MCs, occasionally found in controls along the colonic submucosal vessels and rarely within the *tunica muscularis*, were significantly increased in density in the overall colonic walls of DNBS-treated rats (Figure 4b). Eosinophil density rose significantly throughout the inflamed colonic wall, as compared with the low number of eosinophils detected at mucosal and submucosal levels in controls (Figure 5a). Several eosinophils (Figure 5b, arrows) and macrophages expressing MHC-II antigen (Figure 6, red arrows) were found in close proximity with SP-immunostained fibers in the *lamina propria* and *tunica muscularis* (Figure 5b and Figure 6, green arrows), where the SP-fibers were upregulated and mainly localized in the myenteric ganglia and circular layer of DNBS colon, as compared to controls (Figure 5b).

### 3.7. Evaluation of Glial Activation in the Spinal Cord

Fourteen days after DNBS injection, the morphologies and the numbers of astrocytes and microglial cells were evaluated at the spinal level by an immunofluorescent analysis on lumbo-spinal sections (Figure 7). Both the dorsal and ventral horns of DNBS treated animals were characterized by glial cell activation. Microglia (Iba1-positive cells) did not change in density but underwent well-defined morphological alterations (Figure 7a,c). The percentage of morphologically-activated Iba-1 positive cells was significantly higher in DNBS-treated animals, as compared to controls. The activated status of microglial cells was recognized by the loss of the processes that are peculiar of resting conditions [40]. Astrocytes (GFAP-positive cells) increased significantly in density (about 30%); further a higher number of cells displaying expansions of cellular bodies and processes were given typical activated status [38] in DNBS-treated animals in comparison to control (Figure 7b,d). More details concerning the total number of counted cells were added in Appendix A.

## 4. Discussion

The present study provides a comprehensive pharmacological and histological characterization of persistent visceral pain associated with intestinal inflammatory damage. This kind of chronic pain turned out to be correlated with a persistent immune activation in the proximity of nerve terminals. Interestingly, these alterations, which likely originate in the periphery, reverberate at the central level. Thereby triggering mechanisms of chronicization such as a marked activation of microglia and astrocytes.

The similarities between the animal model of post-inflammatory visceral pain induced by DNBS and the clinical manifestations displayed by patients affected by both IBS and IBDs [5,13,41,42] prompted us to look deeper into the pathophysiological mechanisms involved in pain development and maintenance. DNBS animals developed a long-lasting visceral hypersensitivity to both colo-rectal and rectal stimuli, as observed in patients affected by IBS [43,44]. After the resolution of colitis elicited by DNBS, the animals developed visceral hypersensitivity, which persisted approximately for 90 days, in accordance to previous reports [12]. The evaluation of the magnitude of the abdominal contraction to colo-rectal distension by EMG (VMR) allowed a direct quantitative measure of visceral sensitivity [23], though it needed sleeping animals, thereby excluding the cognitive and emotional components of pain. We completed the puzzle by coupling this test with the measure of AWR to colo-rectal distension in awake animals [25]. The higher sensitivity of AWR allowed us to highlight the long-lasting persistence of visceral hypersensitivity and enabled the analysis of the composite features of this type of pain. With regard for the emotional sphere, we observed that DNBS-treated animals, along with pain, presented also persistent depressive-like behaviors. A high degree of comorbidity of chronic pain with stress-related psychiatric disorders including anxiety and depression occurs as well at the clinical level [45,46]. Mood disorders can be caused by the presence of pain in animals. On the other hand, long term stress facilitates pain perception and sensitizes pain pathways, leading to a feed-forward cycle promoting chronic visceral pain disorders such as IBS [42]. In this context, the benefit provided by antidepressants in patients is justified [7].

From a pharmacological standpoint, we observed that drugs endowed with gut antispasmodic properties were effective in relieving visceral pain, suggesting a close link between hyper-motility and altered sensitivity. By contrast, neither a non-steroidal nor a steroidal anti-inflammatory drug reduced hypersensitivity, irrespective of the inflammation degree. Although inflammation does not appear to be directly responsible for visceral pain, it represents the driving force for the induction of neuroplasticity, leading to altered motility and nociceptor sensitization [41,47]. Similarly to what usually happens in neuropathic pain conditions [48], anti-depressant and anti-epileptic drugs, able to restore the normal patterns of neuronal activity, were effective in reducing visceral hypersensitivity, highlighting a neuropathic face of abdominal pain.

Morphine showed the best pain-relieving profile, making DNBS-treated animals completely insensitive to colo-rectal stimulation—an effect likely explained by the peripheral and central analgesic effects and by the opioid-dependent decrease in bowel motility. The spinal antinociceptive effect of morphine (evaluated by intrathecal infusion) seems to be less relevant in the relief of visceral pain in respect to its affective, supraspinal (limbic) [49] modulation, as attested by the higher efficacy showed by morphine in conscious animals (AWR). Accordingly, intrathecal amitriptyline, by decreasing the affective component of pain, was effective in conscious animals. Pregabalin, an anticonvulsant particularly active against the sensory component of pain, was instead more active (intrathecally) in unconscious animals. Overall, the pharmacological data support the multifaceted (inflammatory, neuropathic, and emotional) nature of this kind of pain.

In ex vivo investigations, we observed an enhanced production and deposition of collagen fibers in the colon of DNBS-treated animals, as previously reported [37]. This process, which starts as a mechanism of repair, unavoidably alters the tissue physiology [50]. On the other hand, we found a widespread increase in the production of substance P, a neurotransmitter involved in the regulation of local nervous signaling and in the modulation of painful perception [51].

During inflammation, resident and immune cells are recruited to the damaged site, where they release inflammatory mediators (i.e., histamine and proteases) able to modify peripheral neuronal activities [52,53,54] and spinal cord transmission, modulating pain maintenance [55]. There are several studies demonstrating bidirectional signaling between MCs and neurons in the human gut [56,57,58]. Histamine and other mediators produced by MCs were found to be increased in the guts of patients affected by IBS, where they play a key role in visceral sensitivity [59], owing to their ability to directly enhance neuron excitability [53,54,60,61]. Similarly, we observed an increased number of MCs in the bowels of DNBS animals, suggesting the persistence of an immune activation during the resolution of the intestinal damage, which unavoidably affects the enteric nerve-signaling [62].

In line with the hypothesis of a persistent immune activation, DNBS rats showed an increase in eosinophils and MHC-II-positive macrophages throughout the colonic wall. Of interest, several of them were found to be in the proximity of SP-positive fibers, suggesting a persistent neuro–immune interaction consequent to colitis. A close apposition of eosinophils to enteric nerves has been reported also in IBD patients [63]. Besides, a pivotal role of activated intestinal macrophages has been suggested in the immune-mediated pathogenesis of functional disturbances both in experimental and clinical IBD [64,65], as well as in enteric neuronal aging [35] and parasitic infections [66]. Of note, both the two latter conditions are characterized by a close contiguity of activated macrophages with intestinal dystrophic neurites.

In addition to immune players, also glial cells, surrounding the somata of sensory neurons, can react to injury and take part to local neuroinflammatory processes [37,52]. Bowel inflammation elicited by DNBS seems to alter the functionality of the enteric nervous system by affecting the interactions among enteric neurons, nociceptors, and enteric glia. In this context, the activation of glia seems to be relevantly involved in the pathophysiology of visceral hypersensitivity [52]. In a previous work conducted on the same model, a persistent activation of enteric glia induced by colitis was demonstrated [37]. Of note, the activation of glial cells seems to spread vertically from the gut to the dorsal root ganglions, where an increased coupling between satellite cells and neurons has been correlated with visceral pain [67].

Like in the periphery, in the central nervous system neurons are surrounded by different types of glial cells, which can be activated by the release of excitatory neurotransmitters [68]. Microglia and astrocytes, particularly, play a strategic role in the central sensitization and remodeling of synaptic plasticity responsible for the chronicization of painful conditions [38,69,70]. The sensitive incoming information is processed at the spinal level by complex circuits in the dorsal horn, and then transmitted to projection neurons for relaying to several brain areas [71,72]. In addition, nociceptive information is conveyed to the ventral horn and contributes to spinally-mediated nocifensive reflexes [71,73]. In DNBS animals, we observed a significant activation of both microglia and astrocytes in the dorsal horn and ventral horn, indicating a diffuse activation of glia over the spinal cord. Of note, the same feature has been detected previously in other pain syndromes associated with hyperreflexia, such as nerve injury [74]. The motor neurons of ventral horn project to abdominal muscles and intestine [74], and therefore the over-activation of astrocytes and microglia in this district could be responsible for the increased VMR to colo-rectal distension. Our results, showing the activation of microglia and astrocytes along the reflex arc of the spinal cord, support the hypothesis that the glia may contribute to prolong the condition of bowel dysregulation and pain, as previously proposed for the animal model of chronic fatigue syndrome [69,75].

In conclusion, the present study highlights the complex nature of the persistent visceral pain caused by colitis in rats and suggests the possibility of employing complementary therapeutic approaches for relieving pain in patients. In this respect, understanding of the molecular mechanisms underpinning the interaction between the periphery and central nervous system, in the framework of this hyper-activated pathophysiological context, could be the way to identify novel therapeutic targets.

## Figures and Tables

**Figure 1 cells-09-01772-f001:**
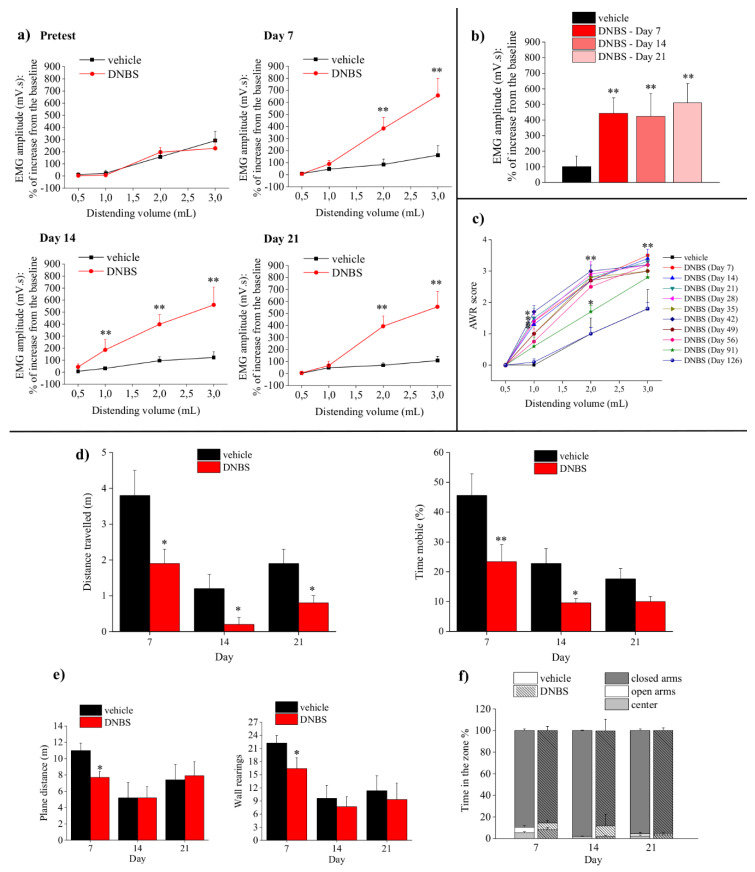
Assessments of visceral hypersensitivity and related behavioral disorders at different time points after DNBS injection. (**a**–**c**) Assessments of visceral sensitivity. (**a**) Viscero-motor response to colo-rectal distension. (**b**) Viscero-motor response to rectal distension. (**c**) Abdominal withdrawal reflex in response to colo-rectal distension. Each value represents the mean ± SEM of 8 animals per group. * *p* < 0.05 and ** *p* < 0.01 vs. vehicle-treated control animals. (**d**–**f**) Assessments of depressive- and anxiety-related behaviors. (**d**) Forced Swim Test; (**e**) Open Field Test; (**f**) Elevated Plus Maze Test. Each value represents the mean ± SEM of 8 animals per group. * *p* < 0.05 and ** *p* < 0.01 vs. vehicle-treated animals.

**Figure 2 cells-09-01772-f002:**
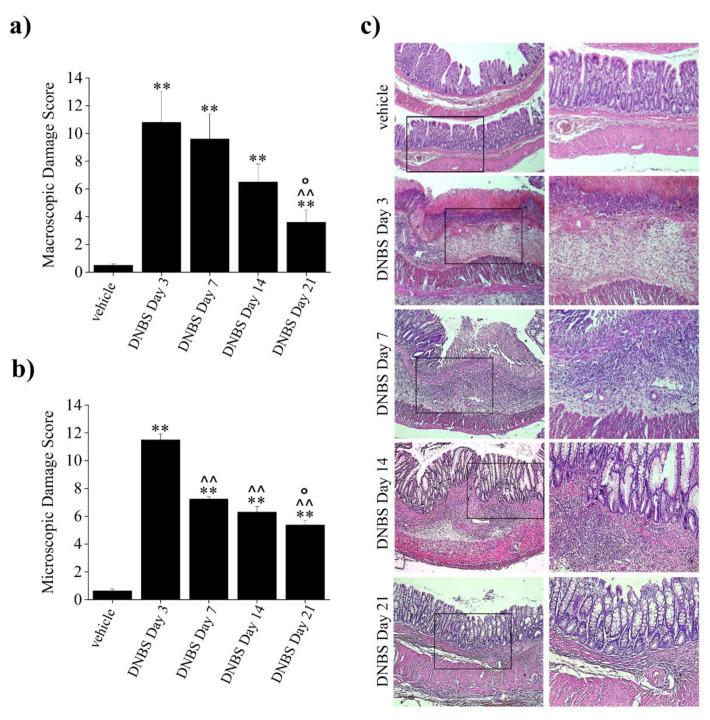
Histological evaluation of colonic damage at different time points after DNBS injection. (**a**) Macroscopic damage score; (**b**) microscopic damage score; (**c**) representative pictures of hematoxylin/eosin-stained sections of full-thickness colon. Original magnification: 5× (left column), 10× (right column). Each value represents the mean ± SEM of 8 animals per group. ** *p* < 0.01 vs. vehicle-treated animals. ^^ *p* < 0.01 vs. DNBS day 3 treated animals. ° *p* < 0.05 vs. DNBS day 7 treated animals.

**Figure 3 cells-09-01772-f003:**
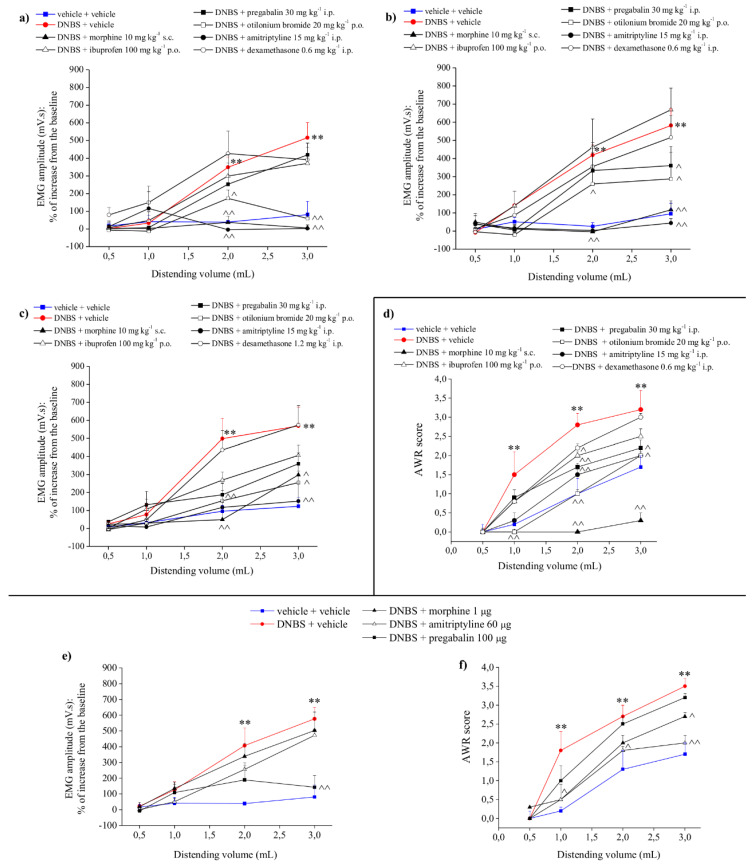
Effects of reference drugs on visceral hypersensitivity; comparison between systemic and intrathecal administration. Systemic effects: (**a**–**c**) viscero-motor response to colo-rectal distension. The tests were performed (**a**) 7, (**b**) 14, and (**c**) 21 days after DNBS injection. Each drug was administered i.p. 15 min before starting the test; (**d**) abdominal withdrawal reflex in response to colo-rectal distension. Tests were performed 14 days after DNBS injection. Each drug was administered i.p. 15 min before starting the test. Intrathecal effects: efficacy of each drug was evaluated on both (**e**) viscero-motor response and (**f**) abdominal withdrawal reflex to colo-rectal distension. The tests were performed 14 days after DNBS injection. Each drug was intrathecally administered 15 min before starting the test. Each value represents the mean ± SEM of 5 animals per group. * *p* <0.05 and ** *p* < 0.01 vs. vehicle + vehicle-treated animals. ^ *p* < 0.05 and ^^ *p* < 0.01 vs. DNBS + vehicle.

**Figure 4 cells-09-01772-f004:**
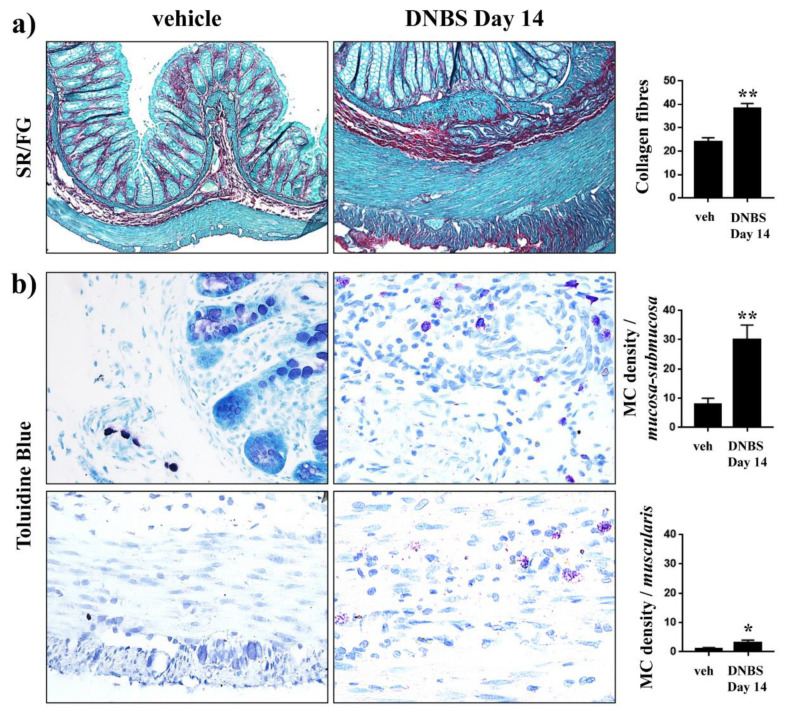
Histological evaluation of colonic fibrosis and mast cell (MC) infiltration 14 days after DNBS injection. Representative pictures of (**a**) Sirius red-fast green (SR/FG)-stained and (**b**) toluidine blue-stained colonic sections obtained from controls or animals treated with DNBS at day 14. For each animal, 5 randomly selected microscopic fields from 2 non-adjacent sections were evaluated. The total numbers of counted MCs were 3–11 (vehicle) and 24–39 (DNBS)/0.41 mm^2^ in the *tunica mucosa-submucosa* and 0–1 (vehicle) and 1–4 (DNBS)/0.41 mm^2^ in the *tunica muscularis*. Column graphs display the mean values of the positive pixels percentages (PPP) of SR-stained collagen fibers, and the density of toluidine blue-stained MCs per mm^2^ of colonic areas (cells/mm^2^) ± S.E.M. obtained from 4 animals for each group. * *p* ≤ 0.05, ** *p* ≤ 0.01 vs. vehicle-treated animals. Original magnification: 20× (**a**), 100× (**b**).

**Figure 5 cells-09-01772-f005:**
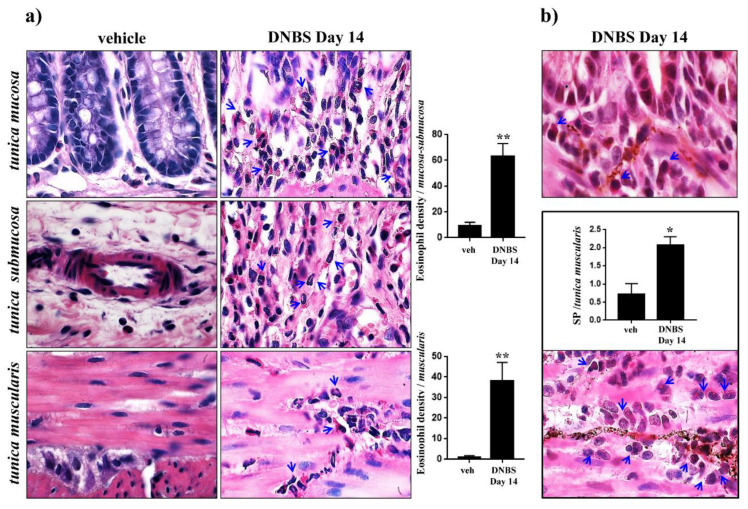
Eosinophil infiltration of colonic wall in close proximity to SP-positive fibers 14 days after DNBS injection. Representative pictures of colonic sections stained with: (**a**) hematoxylin/eosin (eosinophils, arrows) and (**b**) hematoxylin/eosin and SP immunoperoxidase. The right panel (**b**) shows pictures captured from *tunica mucosa* (up) and *muscularis* (down). For each animal, 5 randomly selected microscopic fields from 2 non-adjacent sections were evaluated. The total numbers of counted eosinophils were 5–18 (vehicle) and 42–88 (DNBS)/0.41 mm^2^ in the *tunica mucosa-submucosa* and 1–2 (vehicle) and 20–61 (DNBS)/0.41 mm^2^ in the *tunica muscularis*. Column graphs display the mean values of eosinophil density per mm^2^ of colonic wall areas (cells/mm^2^) (**a**), and positive pixels percentage (PPP) of SP-reactive fibers (**b**) ± S.E.M. obtained from 4 animals for each group. * *p* ≤ 0.05, ** *p* ≤ 0.01, vs. vehicle-treated animals. Original magnification: 100×.

**Figure 6 cells-09-01772-f006:**
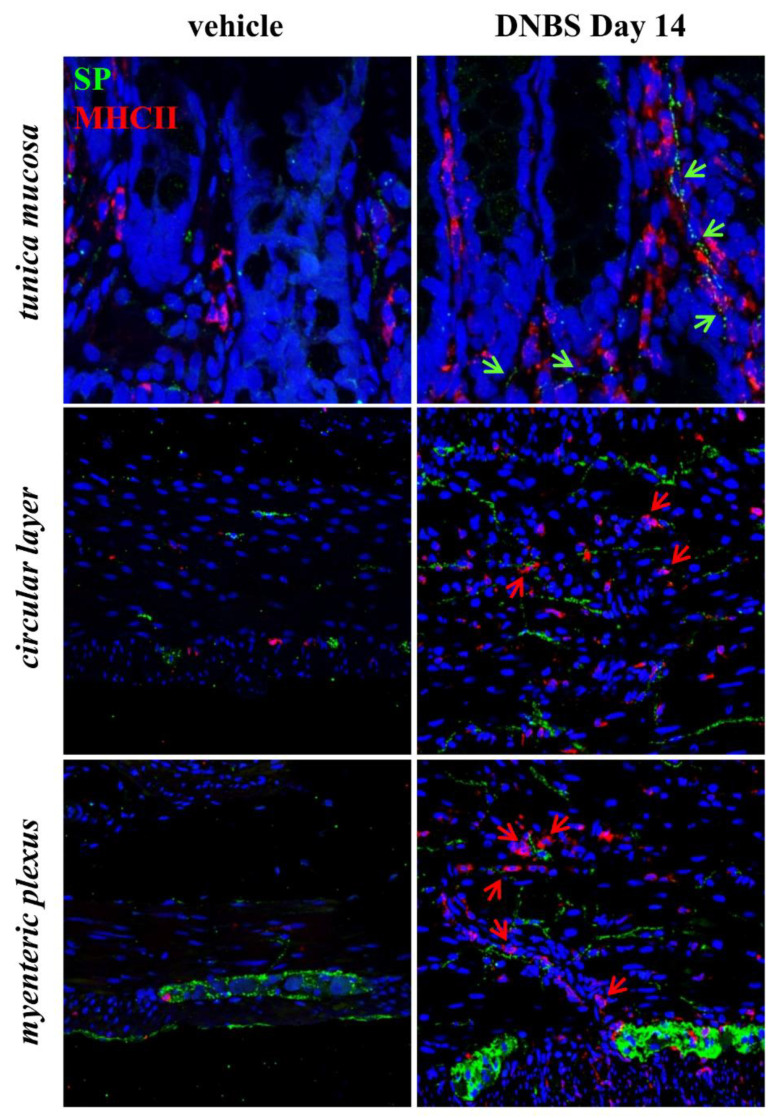
MHC-II cell infiltration of the colonic wall along with SP-immunostained fibers 14 days after DNBS injection. Confocal microscopy representative images of SP (green)/MCHII (red) double-immunolabelled colon from control rats and DNBS-treated rats (*n* = 4 animals for each group). SP-immunostained fibers and MHCII-positive cells are highlighted by green and red arrows, respectively. Original magnification: 80× (*tunica mucosa*), 40× (*tunica muscularis*).

**Figure 7 cells-09-01772-f007:**
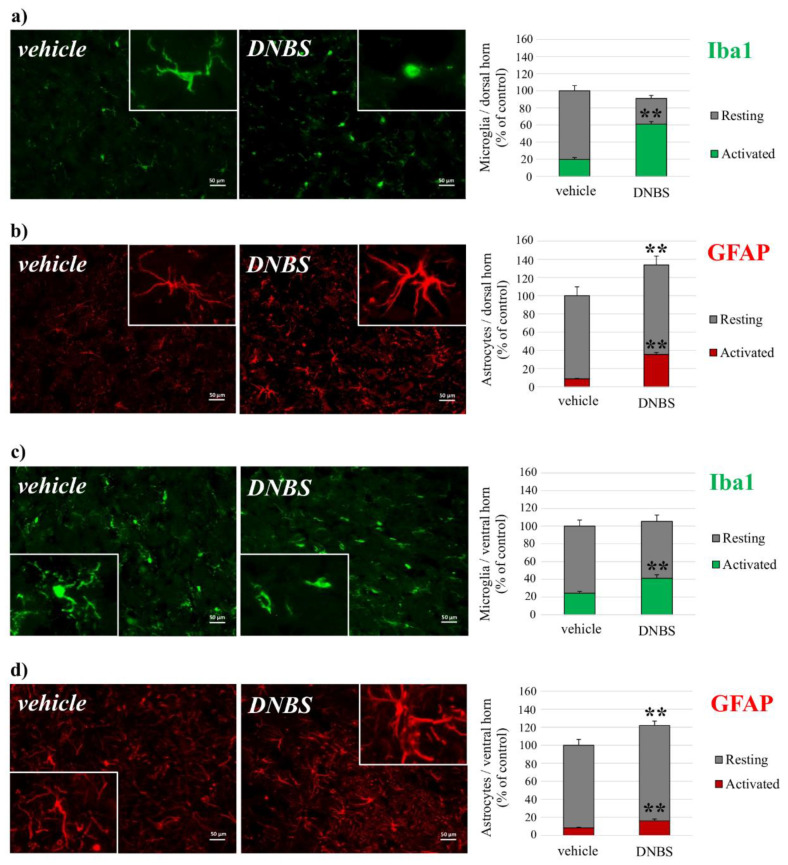
Evaluation of microglia and astrocytes activation in spinal cord horns. Iba1-positive cell density in the (**a**) dorsal and (**c**) ventral horns of the spinal cord 14 days after DNBS injection. Representative images of merged Iba1-labeled microglia cells (green), scale bar: 50 μm; GFAP-positive cell density in the (**b**) dorsal and (**d**) ventral horn of the spinal cords 14 days after DNBS injection. Representative images of merged GFAP-labeled astrocyte cells (red), scale bar: 50 μm. Each value represents the mean of 5 rats, performed by analyzing 8 independent fields for both the sides of the spinal cord. ** *p* < 0.01 vs. control animals.

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
