# Peer review of "Deepening the Mechanisms of Visceral Pain Persistence: An Evaluation of the Gut-Spinal Cord Relationship"

_cells, 2020, doi:10.3390/cells9081772_

Round 1

Reviewer 1 Report

In this study, Lucarini et al. characterize a model of visceral pain induced by rectal administration of 2,4,-dinitrobenzensulfonic acid (DNBS) in rats. The study is interesting and in the scope of Cells. However, I have several concerns:

Major:

Line 97: Please explain the method of intrathecal drug injection in more detail. Did the authors perform a direct lumbar puncture or was a catheter implanted? Which spinal cord level was used?

Sections 2.6 and 2.7: Please provide more details about the histology methods. How was the tissue collected and prepared for histology or immunostaining (fixation? cryosections or paraffin? thickness? etc.)? Which controls have been included to ensure specificity of the stainings?

Line 133: Is the toluidine blue and HE staining specific for mast cells and eosinophils or are other cells also detected?

Line 135 and 141: How long were the primary and secondary antibodies incubated?

Line 146: Which spinal cord level was used?

Line 151: Pelase provide more details of the quantitative analysis of GFAP- and Iba1-positive cells in the spinal cord. How was the specificity of the immunostainings confirmed and the background level determined? Were control stainings without primary antibody included? How was the appropriate spinal cord level confirmed?

The letter size in Figure 1 and 3 is too small.

Figure 1: In Fig. 1d, e and f, the behavior of the two groups before DNBS or vehicle delivery (pretest, like in Fig. 1a) should be included to demonstrate similar baseline behavior in both groups. Moreover, it would be better to use a consistent color for vehicle (e.g. blue) and DNBS (e.g. red) in all parts of Fig. 1 to easier the understanding.

Line 226: Please explain in more detail why day 14 was considered as the optimal time point for further histological analyses. Moreover, although the authors state that day 14 was used for subsequent behavioural analyses, the present day 7, 14 and 21 in Fig. 3.

Figure 3: The overall data presentation is poor. Mean and SEM of the different groups (8 groups per diagram) are badly recognizable and distinguishable. Moreover, wouldn’t a repeated measures ANOVA be more appropriate for statistical analysis?

Line 238-245: Please explain why these doses of morphine, amitriptyline, otilonium bromide, pregabalin, ibuprofen and dexamethasone have been used. Control experiments are needed to ensure that the selected doses do not affect the EMG amplitude in naive or vehicle-treated mice.

Line 239: Why did the authors use morphine at a dose range (5-10 mg/kg) and not at a constant dose? In particular, the fact that morphine completely abolished the DNBS-induced hypersensitivity (Fig. 3d) might indicate that the selected dose was too high.

Line 257-262: Again, a justification for the doses selected for intrathecal delivery is missing.

Figure 4, 5 and 7: How many slides/tissue sections per animal have been used for the quantitative analysis? The total number of cells counted might be added.

Minor:

There are some typos throughout the manuscript that should be corrected. For example, „patients complains“ in line 52 or the dot in line 55.

Reviewer 2 Report

The study by Lucarini et al entitled ‘Deepening the mechanisms of post-inflammatory visceral pain persistence: an evaluation of the gut spinal cord relationship’ was designed to test the efficacy of a range of pharmacological targets on experimentally induced chronic visceral pain.

Visceral pain is a major clinical problem effecting a significant number of people all over the globe. Understanding the mechanisms responsible for the establishment of chronic visceral pain will likely lead to the development of novel therapeutic targets. The authors provide evidence of altered VMR indicative of heightened colonic sensitivity and altered behavioural responses, as well as an increase in depressive like behaviour. However, these observations have previously been published by a number of researchers over the last 2 decades using a range of inflammatory stimuli, including DNBS, and the structurally similar TNBS. As such, there is very little in this manuscript which is novel, as many of the pharmacological treatments that are used in this study have also been previously evaluated in similar animal models.

There are currently a number of major issues with this manuscript which preclude it’s publication that I will outline below.

Overall message

The title of the manuscript is ‘Deepening the mechanisms of post-inflammatory visceral pain persistence: an evaluation of the gut spinal cord relationship’ which implies that the authors have an animal model of chronic pain which follows the resolution of inflammation. However, Figure 2, 4, and 5 indicate quite clearly that this is not a ‘post-inflammatory’ model with significant macro- and microscopic damage in the colon still present, as well as increased mast cell and eosinophil density. This is also stated in the first lines (line 328-329) of the discussion. As such, the manuscript needs to be comprehensively rewritten to reflect the status of the animals studied. A number of similar models in mice show that inflammation is resolved completely at day 28 post TNBS, see https://pubmed.ncbi.nlm.nih.gov/19324867/, and it is unclear why the authors, having shown that hypersensitivity, as indicated by the AWR score, persists indefinitely did not choose a later timepoint for further investigation, e.g. day 91, or day 126.

Methods

Drug administrations – at what level of the spinal cord were i.t injections performed, were the authors targeting pelvic and/or splanchnic sensory innervation pathways?

Assessment of VMR – Increased methodological detail is needed here. There is no description of the surgery to implant electrodes. It is unclear why two different balloon methodologies were used and which was used for the data throughout the manuscript. This is absolutely key as the pelvic and splanchnic pathways provide different types of input to the spinal cord for processing of peripheral stimuli.

How quickly was the balloon distended, over how many seconds? What level of intracolonic pressure was achieved? Does this correlate with physiological and/or painful stimuli?

AWR methodologies need to be expanded. Including if experiments performed in a blinded fashion?

Line 146-147 – ‘Performed by standard protocols, as described previously’ – please provide more detail on these methods.

Which region of the spinal cord was taken for immunofluorescence, e.g. lumbosacral, thoracolumbar?

Statistics – It is not clear if the same rats were used on multiple experimental days or each experimental day reflects a new cohort of animals. These details should be clearly described for all experiments.

Results

All histograms should include individual data points to increase data transparency.

Fig 1: It appears that there are significant differences in all behavioural tests for different time points (Fig 1d-f). These data should be analysed and discussed. Is this a behavioural adaptation to the tests?

Fig 7: Which area of the spinal cord are these images taken from? Are they from areas of the spinal cord responsible for colonic innervation?

Discussion

The discussion needs a compressive re-write to reflect the presence of inflammation in the colon at the experimental time point of day 14.

Line 331 – the authors state that the chronic pain in their study is related to abnormal nerve signalling. A number of studies have shown that colonic afferent hypersensitivity occurs with an increase in the VMR, however, the authors provide no measures of nerve signalling.

Line 346 – No changes in anxiety like behaviour were observed, please discuss why this might have occurred in your model when others have clearly shown increased anxiety behaviours.

Line 352 – 354 – it is implied that anti-spasmodic drugs were effective in relieving pain suggesting a close link between hyper-motility and altered sensitivity. This is pure speculation, as the authors did not measure motility in any assays.

Line 355-357 – you have significant and concurrent inflammation in your experimental model at day 14, how can you then say ‘inflammation does not appear to be responsible for visceral pain’?

Line 362-364 –Opioids, whilst known to have an effect on gastrointestinal motility also have significant analgesic effects on tissues throughout the body which are not related to motility. As such, it is not likely that opioid dependent analgesia is dependent on a decrease in bowel motility in this assay.

Line 382-383 – In your model you still have intestinal damage at day 14, when mast cell numbers were counted? The authors need to clarify how this is related to enteric nervous signalling in the context of sensation? The enteric nervous system is responsible for generating motility, whilst the extrinsic afferents that travel in the splanchnic and pelvic nerve are responsible for sensation.

385-386 – There are no results to suggest that that this is 'in particular'. There are many MHC-II positive macrophages which were not in close proximity to SP positive nerve fibers.

Line 394-397 – These sentences do not make sense. There is no obvious relevance of enteric neurons to this study. How are glia therefore relevant in this context?

Line 398 - In your model you still have intestinal damage at day 14 and this cannot be called a remission phase, when glia activation was measured.

Line 399 – You have not looked at DRG in this study. Please clarify.

Line 404-406 – this is too vague, provide additional detail and references.

Round 2

Reviewer 1 Report

All concerns have been properly addressed.

Reviewer 2 Report

The authors have made considerable effort to address my previous review comments. Where changes were not made, the authors have provided an acceptable rebuttal. The manuscript is now suitable for publication.